# Cognitive Impairment and Non-Communicable Diseases in Egypt’s Aging Population: Insights and Implications from the 2021–2022 Pilot of “A Longitudinal Study of Egyptian Healthy Aging” “AL-SEHA”

**DOI:** 10.3390/ijerph21020151

**Published:** 2024-01-29

**Authors:** Sara A. Moustafa, Reem Deif, Nada Gaballah, Mohamed Salama

**Affiliations:** 1Institute of Global Health and Human Ecology, The American University, New Cairo 11835, Egypt; reem.deif@aucegypt.edu (R.D.); mohamed-salama@aucegypt.edu (M.S.); 2Computer Science and Engineering Department, The American University, New Cairo 11835, Egypt; nadaaym@aucegypt.edu; 3Toxicology Department, Faculty of Medicine, Mansoura University, Dakahlia 35516, Egypt; 4Atlantic Senior Fellow of Equity in Brain Health at the Global Brain Health Institute (GBHI), Trinity College Dublin, D02 PN40 Dublin, Ireland

**Keywords:** aging population, geriatric, NCDs, cognitive functions, Egypt, healthy aging

## Abstract

As the global population ages, the prevalence of cognitive impairment among older individuals has been steadily rising. Like many countries, Egypt is grappling with the challenges an aging demographic poses. The global network of longitudinal aging studies, modeled after the US Health and Retirement Study (HRS), includes over 40 countries but lacks representation from the Arab/North African region. The proposed ‘A Longitudinal Study of Egyptian Healthy Aging’ (AL-SEHA) will address this gap by providing data on aging in Egypt, the largest Arab/North African country, shedding light on the intricate relationship between cognitive impairment and non-communicable diseases (NCDs) in Egypt’s aging population between 2021 and 2022. This study took place in five governments in Egypt and recruited 299 participants from a population of 50+. The results of the study are from the pilot stage of the original longitudinal study (AL-SEHA).

## 1. Introduction

As advocated by the World Health Organization (WHO), healthy aging emphasizes nurturing functional ability in older age for an enriched quality of life [1]. This approach integrates intrinsic capacity (IC) and environmental factors, transitioning elder care from disease-centered to proactive, personalized interventions [2], notably in hospitalized older adults. This perspective underscores the essence of functional ability, encompassing health-related attributes that empower older individuals to pursue valued endeavors. It intertwines an individual’s (IC), a composite of physical and mental capabilities, with environmental factors, thereby enhancing the potential for a higher quality of life while reducing societal burdens. Traditional elder care approaches centered on specific disease markers are transitioning towards a longitudinal, proactive model driven by personalized interventions to amplify intrinsic capacity and functional ability, particularly notable in the context of hospitalized older adults [1,2].

The assessment of IC’s indicators predicts health outcomes, exceptionally functional capability. Domains such as locomotion, vitality, cognition, psychological, and sensory elements are pivotal for maintaining IC in older adults [2]. Beard’s study [3] corroborates this, identifying five subfactors within the English Longitudinal Study on Aging (ELSA) that forecast future functioning. However, a universally accepted IC index is yet to emerge, warranting further exploration. The significance of assessing IC becomes evident in its predictive value for health outcomes, explicitly concerning functional capability. The intricate nature of IC indicators raises inquiries about the most pertinent markers for comprehensive evaluation of an individual’s holistic physical and mental condition. A recent Chinese longitudinal study, CHARLS, further substantiates this structural validity through the ELSA approach [3], collectively accentuating the predictive potential of IC decline and its domains within aging populations. However, a universally acknowledged IC index for clinical or research purposes is yet to be established, necessitating further exploration and quantification of IC concepts in varied contexts.

The escalating prevalence of cognitive impairment, especially with age, necessitates a comprehensive exploration of its implications for individuals’ quality of life [4,5,6]. The escalation in cognitive decline, particularly with increasing age and life expectancy, mirrors the mounting challenges facing developed and developing nations. The resultant surge in dementia not only inflicts substantial economic and social burdens on patients and families but also warrants a comprehensive exploration of its prevalence across genders and residential areas. Discrepancies within the existing literature concerning the gender and residential area-based prevalence of cognitive impairment highlight the need for a cohesive understanding of these dimensions [7,8].

Declines in cognitive function, memory, and mental health [6] often accompany aging. Effective interventions to prevent or delay these declines and improve the quality of life of older adults are crucial. There is a shortage of data on mental health and cognitive functions in the Arab world, particularly in Egypt. Existing studies are limited in scope and do not provide a comprehensive understanding of the issue [6].

Prevalence of mental health issues among older adults in Egypt: Several studies have reported high rates of depression, loneliness, and anxiety among older adults in Egypt [4,7,8]. One study found that 44.4% of participants reported depression [5], while another found that 62.7% of participants were depressed. Loneliness is also a significant problem, with one study finding that 72% of participants experienced severe loneliness [9].

Several factors have been associated with mental health issues among older adults in Egypt, including female gender living in rural areas [8,9] and low levels of education [10]. Social support can also play a protective role, with one study finding that older adults with moderate social support were less likely to experience depression and loneliness [11].

In terms of current needs, Egypt urgently needs a national assessment of the deficiencies of its mental health system. This assessment would provide evidence and reliable information on current needs and barriers to mental health care access. The results of such an assessment could be used to develop locally and culturally appropriate solutions to close the mental health gap.

Socioeconomic factors intricately interplay with cognitive health, with no education higher than high school education, job insecurity, and lifestyle choices influencing susceptibility to cognitive impairments [10]. Notably, individuals with no higher education than high school, diminished IQ, and tenuous job security exhibit a higher susceptibility to cognitive impairments. Furthermore, lifestyle choices play a pivotal role, with tobacco and alcohol usage surfacing as significant predictors of cognitive decline. These connections emphasize the multifaceted nature of cognitive health, which is woven intricately with socioeconomic dimensions and behavioral patterns.

As individuals age, the prevalence of non-communicable diseases (NCDs) increases [4,12]. Combating NCDs is widely acknowledged as essential for promoting healthier aging and ensuring the sustainability of health and social policies, in line with recommendations from the World Health Organization (WHO), the National Institute of Aging (NIA), and the Centers for Disease Control and Prevention (CDC).

NCDs, malnutrition, and depression are prevalent among Egypt’s older population, particularly among women and those with less education [5]. A study conducted at Suez Canal University Hospital [13] in 2020 found that 73% of the 285 older adults with chronic diseases had three or more comorbidities. The most common diagnoses were diabetes mellitus (62.3%), hypertension (75.8%), musculoskeletal pain (61.3%), and eye diseases (51.7%). Additionally, 50.9% and 49.1% of the study participants were independent and partially dependent, respectively, indicating the prevalence of functional disability among older adults with NCDs. It is essential to differentiate between chronic diseases and NCDs; while all NCDs are chronic diseases, not all chronic diseases are NCDs. Some chronic diseases, such as infectious diseases, are communicable.

Cardiovascular Disease (CVD) is another prevalent NCD among older adults in Egypt. A study involving 1661 subjects from across Egypt found that ischemic heart disease was the most common cause of heart failure (HF). Type 2 diabetes mellitus is also a significant concern among Egypt’s older population. A study of 207 Egyptians aged 60 and over found that the mean glycated hemoglobin (HbA1c) level was 8.34 ± 1.09%, indicating poor glycemic control [14]. Only 14% of the participants had an HbA1c level below 7.5%, the recommended target for diabetic patients [14]. As the elderly population grows, the prevalence of NCDs and other health issues highlights the critical need for preventive, curative, and rehabilitative services tailored to this age group. Research on effective prevention, diagnosis, and management of these diseases in older adults is crucial to improving their health outcomes, aligning with the global strategic areas of aging.

Factors such as unemployment and divorce are also linked to health problems in the elderly [12]. NCDs and geriatric depression may increase the risk of malnutrition [13]. Similarly, the presence of NCDs is strongly associated with depression [15,16].

The rapid aging of populations in Low- and Middle-Income Countries (LMICs), particularly in Arabic LMICs with unique social and demographic dynamics, presents a complex set of challenges. The dearth of data on aging in Arabic LMICs hinders effective policymaking and resource allocation. AL-SEHA seeks to bridge this knowledge gap by providing comprehensive data on aging in Egypt, enabling evidence-based policy decisions to navigate the complexities of population aging and foster successful outcomes for older adults in Arabic LMICs [17]. As people age, they are more likely to experience chronic non-communicable conditions. Based on the recommendations of WHO, the National Institute of Aging (NIA), and the Centers for Disease Control and Prevention (CDC), fighting non-communicable diseases is broadly agreed to be the key to health gains at an older age and to making health and social policies sustainable. NCDs, malnutrition, and depression are common in our elderly population, particularly among women and the under-educated [18,19].

Cognitive function, memory, and mental health declines often accompany aging. Research on effective interventions to prevent or delay these declines and improve the quality of life of older adults is critical [20]. With a prevalence of depression, a sense of isolation, and declined cognitive functions in the Egyptian elderly [12,13]. In older people, cognitive impairment is a strong predictor of functional disability and the need for care. Mild cognitive impairment increases the risk of developing dementia, and available evidence suggests that a five-year delay in the age of onset would reduce dementia prevalence by half [14,15,16]. There is limited data on mental health and cognitive functions in the Arab world, specifically in Egypt, where data are specific to certain areas or specific clinical populations rather than those derived from extensive nationwide studies. With this age group expected to grow by 2050, the consequences of dementia will grow as well.

Previous research on cognitive impairment has often been limited to its association with specific chronic diseases, neglecting a comprehensive understanding of cognitive health among the elderly. This study aims to address this gap by examining the prevalence of cognitive impairment in Egypt’s older population, particularly those with chronic conditions.

## 2. Methods

### 2.1. Research Design

The design of this study, characterized by its cross-sectional nature, is warranted as it represents the initial pilot phase of the longitudinal investigation. Consequently, this phase did not encompass any follow-up assessments. The forthcoming Wave 1 will signify the commencement of the longitudinal study, allowing another level of analysis that could possibly establish causal relationships between the variables studied. The pilot study for the AL-SEHA project commenced in January 2021.

We conducted a pilot study using the Arabic SHARE questionnaire to gauge the feasibility of healthy aging research in Egypt. This involved a 10-day Applied Field Work phase, with field implementation supported by coordinators and the AUC team to ensure quality control.

### 2.2. Study Sample

Recognizing the need for an Egyptian aging study modeled after the HRS network, we launched AL-SEHA. This longitudinal survey utilizes the adapted SHARE questionnaire to target non-institutionalized individuals aged 50 and above (excluding those in care facilities). Notably, all eligible individuals within the chosen households were included. Demographic distribution of respondents is illustrated in Table 1.

### 2.3. Inclusion Criteria

Eligibility criteria included the ability to communicate and live independently (at home, rented accommodation, hostels, or retirement homes). Individuals with severe psychiatric issues were excluded. A non-probability convenience sampling approach based on these criteria yielded a final sample size of 299 participants.

### 2.4. Teams Recruited for Fieldwork

We employed a non-probability sample, collecting data from 299 participants aged 50 and above across five governorates with five teams from five Egyptian Universities (Cairo, Beni Suef, Mansoura, Menofia, Suez Canal). All team members were academics from different backgrounds (physicians, geriatricians, pharmacists, and nursing professionals). Each team consisted of a team leader and five members with at least a bachelor’s degree. Data collection instruments included household and individual questionnaires administered via Computer-Assisted Personal Interview (CAPI) and self-completion forms. Quality control measures were implemented concurrently throughout data collection, with CAPI technology further minimizing discrepancies. The cleaned and cross-checked data set, prepared using SPSS, will be used for future analysis and capacity-building workshops.

### 2.5. Fieldwork and Data Management

Investigators from five universities received training on data collection using the Arabic-translated SHARE form; training was on survey methodology, fieldwork, data collection, and data analysis. Eligible older population individuals were carefully selected, and home visits were conducted to interview 15 participants per investigator. Data collection spanned approximately a month and was uploaded to the AUC’s Social Research Center server.

### 2.6. Ethical Considerations and Research Design

The study received approval from the American University in Cairo’s IRB (Case# 2021-2022-029) and obtained written informed consent from all participants. We employed a cross-sectional design, allowing for simultaneous assessment of both dependent and independent variables.

### 2.7. Instruments/Questionnaires

#### 2.7.1. Dependent Variable

Cognitive performance was assessed using a composite measure based on three tests: immediate recall, delayed recall, and verbal fluency. The immediate and delayed recall tasks evaluated short-term verbal learning and memory. Verbal fluency measures executive function and language ability. The composite score ranged from 0 to 1, where 0 to 0.5 indicated impairment and 0.5 to 1 indicated unimpaired cognitive function.

#### 2.7.2. Cognitive Performance Measures

##### Verbal Fluency

Verbal fluency was evaluated by asking participants to name as many animals as possible in one minute. This task assesses executive function, which is responsible for planning, organizing, and carrying out tasks.

##### Immediate Word Recall

Immediate word recall was measured through the Ten-Word Immediate Recall Test. In this test, participants are presented with a list of ten words and asked to recall them immediately. This task evaluates short-term verbal memory, which is the ability to hold information in the mind for a short period of time.

##### Delayed Word Recall

Delayed word recall was assessed using the Ten-Word Delayed Recall Test. Five minutes after the immediate recall test, participants are asked to recall the same ten words. This task evaluates short-term verbal memory, specifically the ability to store and retrieve information over a short delay period.

##### Orientation in Time

Orientation in time was assessed by asking participants to state the current day of the week, month, and year. This task evaluates basic cognitive functioning and orientation to time.

##### Word-List Learning

Word-list learning was measured using the Ten-Word Learning Test. Participants are presented with a list of ten words and asked to recall them immediately, after a five-minute delay, and after a ten-minute delay. This task evaluates verbal learning and memory, specifically the ability to encode, store, and retrieve information over time.

##### Selection of Cognitive Measures

The selection of cognitive measures was guided by the following considerations:Comprehensive evaluation: The measures should provide a comprehensive assessment of various cognitive domains, including memory, executive function, and language ability.Range of difficulty levels: The measures should span a range of difficulty levels to accommodate individuals with varying cognitive abilities.Sensitivity to change: The measures should be sensitive to changes in cognitive functioning over time.Feasibility in survey settings: The measures should be administrable in a survey environment with lay interviewers within a reasonable timeframe.Validity and reliability: The measures should have established validity and reliability in assessing cognitive functioning.Cross-cultural comparability: The measures should use items and instruments that are represented in other national and international surveys to facilitate cross-cultural comparisons.

##### Covariates and Moderators

Cognitive performance in later life is influenced by various factors such as age, education, gender, and marital status. To account for these variables, they were included as covariates in the analyses. Gender was represented as a dummy variable, with 1 for men and 2 for women. Age was also represented as a dummy variable, with 1 for ages between 50 and 60, 2 for ages between 60 and 70, and 3 for ages over 70. Similarly, education was represented as a dummy variable, with 1 for no education, 2 for basic to intermediate education, and 3 for university or postgraduate studies.

##### Statistical Analysis

To begin, descriptive statistics were used to compute. The means and standard deviations of continuous variables as well as the percentages and frequencies of categorical variables.

##### Bivariate Analyses

Bivariate analyses were performed in the second stage to examine the relationship between cognitive health and the study variables, using independent *t*-tests for factors.

## 3. Results

### 3.1. Cognitive Function

The data were extracted from the SPSS (Version 25) dataset for statistical analysis. Bivariate analyses included chi-square tests. As appropriate, data were compared using independent *t*-tests, as presented in Table 2.

### 3.2. Prevalence of Non-Communicable Diseases (NCDs)

Our study focuses on cardiovascular, abdominal, chest, neuro/psychiatric, metabolic, and musculoskeletal diseases and cancer’s impact on cognitive impairment, as presented in Table 3. The prevalence of these diseases varies across age groups and genders, reflecting the intricate web of interactions shaping Egypt’s aging population.

### 3.3. Neuro/Psychiatric Diseases and Cognitive Well-Being

Cognitive impairment reveals the interconnectedness of mental well-being and cognitive faculties. Conditions like insomnia show statistically significant associations with cognitive impairment (*p*-value: 0.0169). The intricate interplay between cognitive well-being and mental health becomes evident when exploring associations between neuropsychiatric diseases and cognitive impairment.

### 3.4. Metabolic Diseases and Their Impact on Cognition

Metabolic diseases, epitomized by diabetes, demonstrate their nuanced relationship with cognitive health through statistically significant associations with cognitive impairment (*p*-value: 0.0137). These findings accentuate the urgency of interventions that holistically address metabolic and cognitive well-being, potentially ameliorating cognitive decline progression.

### 3.5. Respiratory Health and Cognitive Implications

Associations between chest diseases and cognitive impairment underline the intricate connection between respiratory health and cognitive well-being. Conditions like hard breathing and cough show statistically significant associations with cognitive impairment (*p*-values: 0.000076, 0.0032). The intricate correlation between respiratory health and cognitive well-being materializes in associations between chest diseases and cognitive impairment.

### 3.6. Musculoskeletal Health and Functional Independence

The intricate dance between physical well-being and cognitive health manifests as musculoskeletal diseases and cognitive impairment associations. Ailments like falls, fear of falls, and dizziness, often rooted in musculoskeletal health, demonstrate statistically significant associations with cognitive impairment, underscoring the profound interaction between the physical and cognitive domains (*p*-values: 0.00108, 0.001089, 0.0089).

### 3.7. Cancer and Its Nuanced Prevalence

Cancer prevalence in the older Egyptian population was notably low. Specifically, among those aged 50–60 years, only 1.32% had cancer, with 98.68% without the disease. In the 61–70 age group, 3.41% were affected by cancer, while 96.59% were not. Similarly, among individuals aged 71 and above, cancer was found in 3.33%, with 96.67% unaffected. However, larger and more diverse samples are needed to draw more definitive conclusions about cancer rates in the Egyptian older population. Acknowledging these limitations is essential for a balanced perspective on the findings, and further research is required for more accurate insights into the cancer burden among the older population in Egypt.

## 4. Discussion

The findings of this study underscore the urgent need for comprehensive and evidence-based approaches to address the multifaceted challenges faced by Egypt’s aging population, particularly in the context of cognitive impairment and NCDs. Our study highlights the prevalence of NCDs and cognitive impairment among older adults in Egypt, emphasizing the need for effective prevention, diagnosis, and management strategies to mitigate their impact on overall well-being. These findings align with existing research suggesting that NCDs contribute to cognitive decline and dementia [21,22,23].

The intricate interplay between NCDs and cognitive impairment calls for a holistic approach to healthcare that addresses both physical and cognitive well-being [24]. Our study highlights the significant association between NCDs, such as diabetes, hypertension, and cardiovascular diseases, and cognitive impairment [21,22]. These findings emphasize the need for integrated healthcare strategies addressing NCD management and cognitive health promotion. This approach aligns with the World Health Organization’s (WHO) Integrated Chronic Disease Management (ICDM) framework, which promotes a person-centered approach to managing multiple chronic conditions [25,26].

The multifaceted nature of cognitive impairment necessitates a comprehensive assessment encompassing various domains, including locomotion, vitality, cognition, psychological well-being, and sensory function [27,28,29]. Our study’s findings underscore the importance of evaluating these domains to better understand an individual’s cognitive health [30,31,32]. The rising prevalence of cognitive impairment, especially among older adults, demands a concerted effort to identify and address its underlying causes. Our study’s findings shed light on the association between cognitive impairment and socioeconomic factors, such as having no education higher than high school education, job insecurity, and lifestyle choices like tobacco and alcohol use. These findings align with existing research highlighting the interplay between socioeconomic determinants and cognitive health [30,31,32]. Addressing these socioeconomic factors through targeted interventions, such as educational programs, employment support, and lifestyle modification initiatives, can be crucial in promoting cognitive health and reducing the risk of cognitive impairment [33,34]. 

The complex relationship between cognitive impairment and NCDs highlights the need for integrated approaches that address both domains. However, the results of this work cannot be generalized to the Egyptian population since the sample was a convenient, non-random sample. A follow-up study using random sampling is important to validate the current findings.

### 4.1. Healthcare Planning and Resource Allocation

Understanding associations between cognitive impairment and diseases aids in strategic resource allocation for timely interventions. Recognizing associations between cognitive impairment and diverse diseases augments the strategic blueprint for resource allocation within the Egyptian healthcare system. Informed by these associations, policymakers can channel resources towards timely interventions that ameliorate cognitive decline, enhance individual well-being, and alleviate societal burdens.

### 4.2. Catalyzing Research and Collaboration

Associations between cognitive impairment and diseases foster interdisciplinary research and collaboration for holistic strategies. The implications of cognitive impairment’s associations with diseases extend beyond individual interventions, catalyzing collaborative research across disciplines in Egypt and the MENA region. This interdisciplinary endeavor propels the creation of comprehensive strategies that transcend traditional healthcare silos, forging innovative pathways to address the intricate web of cognitive decline.

### 4.3. Conclusion: Enabling a Brighter Aging Future for Egypt

Evidence-based policies and interventions pave the way for a healthier and more inclusive aging demographic. In conclusion, synthesizing evidence-based policies and interventions is the cornerstone of facilitating a brighter, healthier aging trajectory for Egypt. The intricate interplay between cognitive impairment and non-communicable diseases unravels novel avenues for policymaking and interventions, steering the nation toward a future that prioritizes the well-being and dignity of its aging population.

### 4.4. Limitations of the Study and Suggestions for Future Research in the Egyptian Context

Acknowledging limitations, future research can employ longitudinal designs and objective measures to enhance understanding. Admitting the inherent limitations of this study, future research endeavors within the Egyptian context can harness the power of longitudinal designs and accurate measurements. These methodological refinements promise to illuminate deeper insights into the dynamic interplay between cognitive impairment and diseases, thereby enriching the foundation for comprehensive interventions. The results of this work cannot be generalized to the Egyptian population since the sample was a convenient, nonrandom sample. A follow-up study using random sampling is important to validate the current findings.

In conclusion, this paper delves into the intricate web connecting cognitive impairment and NCDs within Egypt’s aging population. We illuminated a tapestry of interactions that transcend traditional healthcare boundaries by investigating associations between cognitive decline and diverse diseases. The implications of these findings resonate through evidence-based policies, cultural considerations, interdisciplinary collaboration, and, most importantly, empowered aging experiences. These insights pave the way for Egypt toward a future characterized by enhanced well-being, dignity, and inclusivity for its elderly citizens.

## Figures and Tables

**Table 1 ijerph-21-00151-t001:** Demographics.

	N	%
Gender		
Male	128	42.81
Female	171	57.19
Educational Attainment		
No education	48	16.05
Primary education	31	10.37
Preparatory education	18	6.02
High school	48	16.05
Medium-high	18	6.02
Bachelor	112	37.46
Postgrad	24	8.03
Age		
50–60	151	50.5
61–70	88	29.43
71–80	50	16.72
81–90	8	2.68
Above 91	2	0.67
Marital Status		
Unmarried	5	1.67
Engaged	0	0
Married	190	63.55
Separated	3	1
Divorced	13	4.35
Widow	88	29.43

**Table 2 ijerph-21-00151-t002:** Demographic characteristics and Chi-squared analysis of impaired and unimpaired groups in a study population.

	Impaired (n = 118)		Unimpaired (n = 181)		Chi-Squared Test	*p*-Value
Age	N	%	N	%		
50–60	52	44.07	99	54.7		
61–70	39	33.05	49	27.07		
71+	27	22.88	33	18.23	3.23	0.2
Mean (SD)	62.338 (8.8087)		60.635 (8.0505)			
Gender						
Male	44	37.29	84	46.41		
Female	74	62.71	97	53.59	2.07	0.15
Education						
No education	29	24.58	19	10.5		
Basic/intermediate	52	44.07	63	34.81		
University	37	31.36	99	54.7	18.968	0.0001
Marital Status						
Single	1	0.85	4	2.21		
Married	69	58.47	121	66.85		
Separated	3	2.54	0	0		
Divorced	3	2.54	10	5.52		
Widow	42	35.59	46	25.41	10.159	0.0378

**Table 3 ijerph-21-00151-t003:** Descriptive statistics for 8 categories of non-communicable diseases classified by age, gender, education, and cognitive functions.

Disease Name		Diseases	No Disease	Percentage	Disease	Percentage %
Cardiovascular Disease: (Heart disease, hypertension, cholesterol, and angina)	Age group	50–60	492	81.46	112	18.54
		61–70	244	69.32	108	30.68
		71 and above	150	62.50	90	37.50
	Gender	Male	371	72.46	141	27.54
		Female	515	75.29	169	24.71
	Education	No education	150	78.13	42	21.88
		Basic/intermediate	335	72.83	125	27.17
		University/post-grad	401	73.71	143	26.29
	Cognition	Unimpaired	563	77.76	161	22.24
		Impaired	323	68.43	149	31.57
Neuro/psychiatric Disease: (Parkinson’s Disease, Insomnia, Fall, Fear of Falling, Dizziness, Brain Clots)	Age group	50–60	834	92.15	71	7.85
		61–70	463	87.69	65	12.31
		71 and above	297	82.50	63	17.50
	Gender	Male	695	90.49	73	9.51
		Female	899	87.71	126	12.29
	Education	No education	254	88.50	33	11.50
		Basic/intermediate	621	90.00	69	10.00
		University/post-grad	719	88.11	97	11.89
	Cognition	Unimpaired	1002	92.35	83	7.65
		Impaired	593	83.88	114	16.12
Metabolic Disease (Diabetes)	Age group	50–60	127	84.11	24	15.89
		61–70	55	62.50	33	37.50
		71 and above	28	46.67	32	53.33
	Gender	Male	91	71.09	37	28.91
		Female	119	69.59	52	30.41
	Education	No education	34	70.83	14	29.17
		Basic/intermediate	77	66.96	38	33.04
		University/post-grad	99	72.79	37	27.21
	Cognition	Unimpaired	137	75.69	44	24.31
		Impaired	73	61.86	45	38.14
Abdominal and Gastrointestinal Disease: (Stomach Ulcer, Urine Incontinence, Bowel Problems)	Age group	50–60	407	89.85	46	10.15
		61–70	215	81.44	49	18.56
		71 and above	148	82.22	32	17.78
	Gender	Male	339	88.28	45	11.72
		Female	431	84.02	82	15.98
	Education	No education	127	88.19	17	11.81
		Basic/intermediate	308	89.28	37	10.72
		University/post-grad	335	82.11	73	17.89
	Cognition	Unimpaired	463	85.27	80	14.73
		Impaired	307	86.72	47	13.28
Chest Disease: (Asthma, Lung Disease, Hard Breathing, Cough)	Age group	50–60	565	93.54	39	6.46
		61–70	314	89.20	38	10.80
		71 and above	222	92.50	18	7.50
	Gender	Male	467	91.21	45	8.79
		Female	634	92.69	50	7.31
	Education	No education	185	96.35	7	3.65
		Basic/intermediate	428	93.04	32	6.96
		University/post-grad	488	89.71	56	10.29
	Cognition	Unimpaired	689	95.17	35	4.83
		Impaired	412	87.29	60	12.71
Cancer	Age group	50–60	149	98.68	2	1.32
		61–70	85	96.59	3	3.41
		71 and above	58	96.67	2	3.33
	Gender	Male	124	96.88	4	3.13
		Female	168	98.25	3	1.75
	Education	No education	47	97.92	1	2.08
		Basic/intermediate	113	98.26	2	1.74
		University/post-grad	132	97.06	4	2.94
	Cognition	Unimpaired	176	97.24	5	2.76
		Impaired	116	98.31	2	1.69
Musculoskeletal Disease: (Osteoporosis, Arthritis, Hip Fracture, Back/Joint Pain)	Age group	50–60	480	79.47	124	20.53
		61–70	260	76.25	81	23.75
		71 and above	161	67.08	79	32.92
	Gender	Male	427	83.40	85	16.60
		Female	474	69.40	209	30.60
	Education	No education	131	68.23	61	31.77
		Basic/intermediate	365	79.52	94	20.48
		University/post-grad	405	74.45	139	25.55
	Cognition	Unimpaired	554	76.52	170	23.48
		Impaired	347	73.67	124	26.33

## Data Availability

The datasets used and analyzed during the current study are available from the corresponding author upon reasonable request.

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
