# Peer review of "Cognitive Impairment and Non-Communicable Diseases in Egypt’s Aging Population: Insights and Implications from the 2021–2022 Pilot of “A Longitudinal Study of Egyptian Healthy Aging” “AL-SEHA”"

_ijerph, 2024, doi:10.3390/ijerph21020151_

Round 1
Reviewer 1 Report
Comments and Suggestions for Authors
1. The abbreviation in the topic: AL-SEHA, what is the A?
2. Please describe the differences between non-communicable diseases and chronic diseases.
3. Healthy aging or just aging in lines 106-107 ?
4. The calculation method of sample size needs to be stated. What is the method to recruit the participants ?
5. The authors need to describe the training process for investigators and the reliability of the investigators.
6. The categories in the variable of Age are overlapping, e.g., 50-60, 60-70, 60 is overlapping.
7. The categories in Table 1 and Table2 are not consistent. The reasons need to be described.
8. The heading of result is missing.
9. The contents of methods and results need to be rearranged, e.g., Table 2 is methods or results?
10. The independent variable needs to be explained.
11. P.7 The results showed in the text cannot be seen in the Table 3.
12. The discussion is beyond the findings of results.
Author Response
Response to reviewers-IJEPH
All changes and modifications in the text are highlighted in red.
We would like to thank the reviewers for their feedback and comments to increase the value of this work!
Reviewer 1:
- The abbreviation in the topic: AL-SEHA, what is the A?
A Longitudinal Study of Egyptian Healthy Aging
- Please describe the differences between non-communicable diseases and chronic diseases.
Done in lines 110-1113
- Healthy aging or just aging in lines 106-107 ?
The whole section is modified, starting from lines 97-113
- The calculation method of sample size needs to be stated. What is the method to recruit the participants?
Elaborated on that in lines 167-218
- The authors need to describe the training process for investigators and the reliability of the investigators.
Elaborated on that in lines 193-213
- The categories in the variable of Age are overlapping, e.g., 50-60, 61-70, 60 is overlapping.
Modified
- The categories in Table 1 and Table2 are not consistent. The reasons need to be described.
Modified
- The heading of result is missing.
Done
- The contents of methods and results need to be rearranged, e.g., Table 2 is methods or results?
Modified
- The independent variable needs to be explained.
Elaborated on it
- The results showed in the text cannot be seen in the Table 3.
Yes, as to keep the tables neat and easier to understand, so we preferred to elaborate in data results intext rather than Table3
- The discussion is beyond the findings of results.
The discussion was rewritten to address this comment.
Reviewer 2 Report
Comments and Suggestions for Authors
Introduction
· WHO reference is needed for the first sentence
· lines 27-28, reword the sentence as it is almost the same as the first one in the introduction section
· Instead of several subheadings, I would put the text under one heading "Introduction".
· line 106, delete the title
Methods
· line 109, upon the first appearance of the acronym HRS you are obliged to put the full name in the parenthesis. After that only abbreviation is necessary. The same applies to the acronym AL-SEHA and SHARE.
· Please, make it clearer on which questionnaire your study is based!
· lines 116-117, merge sentences with the sentence talking about respondents aged 50 and above (lines 110-111)
· the same text is under the research design as in the methods section, so merge the text!
· The text is repeated, the authors are talking about the same thing two or three times under different subheadings. So please structure the method section as follows: research design and study sample, instruments/questionnaires, variables and statistic analyses.
· be more precise about how the cognitive performance score was assessed. It would be clearer if you merge text under the titles "dependent variable" and "cognitive performance"!
Result section should be separated from the methods section. Please, reword the titles of tables 2 and 3. They are not appropriate ones. Be more precise!
The discussion is without references. There are no results of the other studies. Only policy implications and interventions were mentioned.
Comments on the Quality of English LanguageMinor editing of English language required
Author Response
Response to reviewers-IJEPH
All changes and modifications in the text are highlighted in red.
We would like to thank the reviewers for their feedback and comments to increase the value of this work!
Reviewer 2:
Title and Abstract
- Since the study mentioned that the study adopted a longitudinal design, please mention the study’s relevant period in the title and abstract.
This is a cross sectional study, it is the pilot stage of the first wave of the planned longitudinal study, which will start its first wave in the middle of 2024.
- The research gap of the study was presented in the abstract section. Please improve the novelty of the study in the abstract section.
Done
- Please provide the study setting, including the location and relevant study period. The overall method information must be presented.
Done
- The result should be presented as well.
Done
Introduction
- Please improve what makes Egypt different from other countries. I mean, many countries have similar problems, but the study must highlight the scientific issue in Egypt.
Done
- Please discuss the study participants.
Done in lines 188-192
Method
- How did the author determine the study size?
A non-probability convenience sampling approach based on these criteria yielded a final sample size of 299 participants.
- The study adopted a longitudinal study; however, I could not find the follow-up point for data collection.
The design of this study, characterized by its cross-sectional nature, is warranted as it represents the initial, pilot phase of the longitudinal investigation. Consequently, this phase did not encompass any follow-up assessments. The forthcoming Wave 1 will signify the commencement of the longitudinal study which will allow another level of analysis that could possibly establish causal relationships between the variables studied.. The pilot study for the AL-SEHA project commenced in January 2021.
- Who collects the data? What are the criteria?
Elaborated on that in lines 194-214
- In the study title, the author mention “Longitudinal Study of Egyptian Healthy Aging “AL-SEHA”. However, in the research design section, this study employed a cross sectional study. Please make it clear.
The design of this study, characterized by its cross-sectional nature, is warranted as it represents the initial, pilot phase of the longitudinal investigation. Consequently, this phase did not encompass any follow-up assessments. The forthcoming Wave 1 will signify the commencement of the longitudinal study which will allow another level of analysis that could possibly establish causal relationships between the variables studied.. The pilot study for the AL-SEHA project commenced in January 2021. And made it clear in the paper
- Please provide the detail information of setting, participants, potential bias how to deal with it.
Elaborated on that in lines 175-214
- Please provide the information of instrument for each variable, including how to interpret the instrument score.
Elaborated on that in lines 222-288
- I recommend to improve the statistical method to be more advance and increased internal validity. It can include the covariate to be adjusted in the final result.
With due respect, we preferred to focus on analyzing the set of determinants we identified and correlate to healthy aging process.
- How about the study generalizability issue?
The results of this work can not be generalized to the Egyptian population, since the sample was a convenient, non random sample. A follow up study using random sampling is important to validate the current findings

Reviewer 3 Report
Comments and Suggestions for Authors
Cognitive Impairment and Noncommunicable Diseases in Egypt's Aging Population: Insights and Implications from the Pilot of the Longitudinal Study of Egyptian Healthy Aging “AL-SEHA”
Title and Abstract
- Since the study mentioned that the study adopted a longitudinal design, please mention the study’s relevant period in the title and abstract.
- The research gap of the study was presented in the abstract section. Please improve the novelty of the study in the abstract section.
- Please provide the study setting, including the location and relevant study period. The overall method information must be presented.
- The result should be presented as well.
Introduction
- Please improve what makes Egypt different from other countries. I mean, many countries have similar problems, but the study must highlight the scientific issue in Egypt.
- Please discuss the study participants.
Method
- How did the author determine the study size?
- The study adopted a longitudinal study; however, I could not find the follow-up point for data collection.
- Who collects the data? What are the criteria?
- In the study title, the author mention “Longitudinal Study of Egyptian Healthy Aging “AL-SEHA”. However, in the research design section, this study employed a cross sectional study. Please make it clear.
- Please provide the detail information of setting, participants, potential bias how to deal with it.
- Please provide the information of instrument for each variable, including how to interpret the instrument score.
- I recommend to improve the statistical method to be more advance and increased internal validity. It can include the covariate to be adjusted in the final result.
Discussion
- How about the study generalizability issue?
Author Response
Response to reviewers-IJEPH
All changes and modifications in the text are highlighted in red.
We would like to thank the reviewers for their feedback and comments to increase the value of this work!
Reviewer 3:
The increase in the population over 50 years of age is a constant worldwide. Thus, this fact has perhaps become one of the issues and challenges that have more homogenized the world population. This reality requires an effort not only in social policies but, above all, in health policies. In this way, the relationship between physical and mental health takes on special relevance when we analyze what happens as we get older. In this sense, this research work reflects this need to intervene jointly in the physical, functional and mental aspects of older people. This implies the confirmation of the importance of the interaction between cognition and the suffering of certain diseases. Thus, works like this one are necessary to improve the quality of life of older people, in this case, who live in Egypt. The work presented - Cognitive decline and non-communicable diseases in 2 the aging population in Egypt: insights and implications from the 3 pilot of the longitudinal study on Egyptian healthy aging 4 “AL-SEHA” – generally meets the necessary conditions that are required for the publication of a work of this type. Although the theoretical and methodological approach is correct, there are a series of issues that could contribute to improving the work and are detailed below:
- It is advisable not to categorically state aspects such as the following: Notably, individuals with lower educational attainment, diminished IQ, and ten-65 uous job security exhibit higher susceptibility to cognitive impairments. It is important to clarify the relative nature of this relationship. So, we know that there seems to be a relationship, but there is no evidence that this relationship is causal.
Done
- Likewise, it is necessary to be very respectful when making statements of the following type: Aging is often accompanied by cognitive function, memory, and mental health de-86 clines. Aging is not necessarily the same as being sick, nor is it the same as suffering from cognitive decline. There are certain stereotypes towards older people – such as that old age inevitably leads to cognitive decline – that must be eradicated.
Done
2 • On the other hand, it is necessary to include the evaluation instruments used in the research work. Along these lines, it is necessary to explain or show the tasks that have been used to evaluate the variables studied (verbal fluency, immediate word recall and delayed word recall); as well as the neuropsychological tasks used. Regarding this issue, there are tests such as the Mini Mental State Examination (MMSE) (Folstein, 1983), whose reliability and validity to evaluate cognitive functioning have been demonstrated in various research works.
Elaborated on that in lines 222-288
- Finally, as recognized in the limitations section, it is necessary to expand this study using another type of analysis – of a longitudinal nature – that allows establishing causal relationships between the variables studied.
First wave of the AL-SEHA will be launched mid 2024.

Reviewer 4 Report
Comments and Suggestions for Authors
Comments and suggestions for authors are attached in a Word document.

Author Response
Response to reviewers-IJEPH
All changes and modifications in the text are highlighted in red.
We would like to thank the reviewers for their feedback and comments to increase the value of this work!
Reviewer 4:
Introduction · WHO reference is needed for the first sentence ·
Has been added
lines 27-28, reword the sentence as it is almost the same as the first one in the introduction section
Done
- Instead of several subheadings, I would put the text under one heading "Introduction".
Done
line 106, delete the title Methods ·
Done
line 109, upon the first appearance of the acronym HRS you are obliged to put the full name in the parenthesis.
Done
After that only abbreviation is necessary. The same applies to the acronym ALSEHA and SHARE. ·
Please, make it clearer on which questionnaire your study is based!
Done
- lines 116-117, merge sentences with the sentence talking about respondents aged 50 and above (lines 110-111) ·
Done
the same text is under the research design as in the methods section, so merge the text! ·
Done
The text is repeated, the authors are talking about the same thing two or three times under different subheadings. So please structure the method section as follows: research design and study sample, instruments/questionnaires, variables and statistic analyses. · be more precise about how the cognitive performance score was assessed. It would be clearer if you merge text under the titles "dependent variable" and "cognitive performance"!
Done
Result section should be separated from the methods section.
Done
Please, reword the titles of tables 2 and 3. They are not appropriate ones. Be more precise!
Done
The discussion is without references. There are no results of the other studies. Only policy implications and interventions were mentioned. Minor editing of English language required
Done and references were added to the discussion

Round 2
Reviewer 1 Report
Comments and Suggestions for Authors
No comments.
Author Response
The authors would like to thank you for your constructive feedback to improve the quality of our work.
Reviewer 2 Report
Comments and Suggestions for Authors
- introduction, lines 34-35, once again reword the sentence as it is almost the same as the first one in the introduction section
- line 131, AL-SEHA must be without parenthesis
- why the method section has so many subheadings? It looks too complex. It is not structured as follows: research design and study sample, instruments/questionnaires, variables, and statistic analyses.
Minor editing is necessary!
Author Response
We would like to thank the reviewers for their feedback and comments to increase the value of this work!
All changes and modifications in the text are highlighted in red with yellow shading.
- introduction, lines 34-35, once again reword the sentence as it is almost the same as the first one in the introduction section
Sentence deleted to avoid redundancy
- line 131, AL-SEHA must be without parenthesis
Done
why the method section has so many subheadings? It looks too complex. It is not structured as follows: research design and study sample, instruments/questionnaires, variables, and statistic analyses.
- As required by other reviewers, we elaborated on the method section, with explaining each and every instrument used, its rationale and how it was conducted. These subheadings were added to avoid confusion and make it clear as we focus on each and every part of the study design.
- Changed some of the subheadings according to recommendation